# Multiple phosphorylation of the Cdc48/p97 cofactor protein Shp1/p47 occurs upon cell stress in budding yeast

Alexander Agrotis, Frederic Lamoliatte, Thomas D Williams, Ailsa Black, Rhuari Horberry, Adrien Rousseau

The homohexameric p97 complex, composed of Cdc48 subunits in yeast, is a crucial component of protein quality control pathways including ER-associated degradation. The complex acts to segregate protein complexes in an ATP-dependent manner, requiring the engagement of cofactor proteins that determine substrate specificity. The function of different Cdc48 cofactors and how they are regulated remains relatively poorly understood. In this study, we assess the phosphorylation of Cdc48 adaptor proteins, revealing a unique and distinctive phosphorylation pattern of Shp1/p47 that changed in response to TORC1 inhibition. Site-directed mutagenesis confirmed that this pattern corresponded to phosphorylation at residues S108 and S315 of Shp1, with the double-phosphorylated form becoming predominant upon TORC1 inhibition, ER-stress, and oxidative stress. Finally, we assessed candidate kinases and phosphatases responsible for Shp1 phosphorylation and identified two regulators. We found that cells lacking the kinase Mpk1/Slt2 show reduced Shp1 phosphorylation, whereas impaired PP1 phosphatase catalytic subunit (Glc7) activity resulted in increased Shp1 phosphorylation. Overall, these findings identify a phosphoregulation of Shp1 at multiple sites by Mpk1 kinase and PP1 phosphatase upon various stresses.

## Introduction

The ubiquitin–proteasome system (UPS) is crucial for proper cellular function, with its dysfunction implicated in many diseases including cancer and neurodegeneration (1, 2). The central component of the UPS, the proteasome, is a large multisubunit protein complex that acts like a "molecular paper shredder" to proteolytically degrade most of the cells' damaged and short-lived proteins. Proteins are typically targeted to the proteasome by (poly)-ubiquitination—a post-translational modification involving the covalent attachment of one (or multiple) copy of the small protein ubiquitin (1).

A substantial proportion of proteasome substrates cannot be accessed by the proteasome because of their physical state or subcellular localisation. These substrates rely on another essential UPS component known as the p97/VCP complex (or Cdc48 complex in yeast). The Cdc48/p97 complex is a homohexameric AAA+ ATPase ring that acts upstream as a "helping hand" for the proteasome. It functions as a segregase enzyme to disassemble protein aggregates and multiprotein complexes, enabling subsequent proteasomal degradation of their protein constituents (3). It also plays a crucial role in endoplasmic reticulum–associated degradation (ERAD) by providing the pulling force for retrotranslocation of misfolded ER proteins, and thus facilitating their proteasomal degradation (4). In addition, the Cdc48/p97 complex has distinct functions that do not necessarily conclude with substrate proteasomal degradation, including chromatin disassembly and signalling complex disassembly to modulate signalling pathway activation (5, 6).

Diverse functions of the Cdc48/p97 complex are coordinated by interchangeable cofactor proteins, which determine substrate specificity. These cofactors typically possess distinct protein domains, which separately bind Cdc48 and polyubiquitinated substrate proteins (7). Among the most well-characterised cofactors are Npl4 and Ufd1, which form a heterodimer within the Ufd1-Npl1-Cdc48 complex, and together are mainly involved in ERAD and proteasomal degradation (8). In addition, there is a family of Ubx proteins which share a UBX (ubiquitin-like) domain (9), consisting of Ubx1-7 in yeast (10). The best studied of these, Shp1/Ubx1 (known as p47 in mammals), has been implicated alongside Cdc48/p97 in processes such as autophagy, cell cycle progression, and protein complex quality control (5, 11, 12, 13).

Very little is known about how Cdc48/p97 cofactors are regulated by post-translational modifications under different stress conditions. One of the most common post-translational modifications is phosphorylation, where a phosphate molecule is reversibly attached to serine, threonine, or tyrosine residues by the action of kinases (and removed by phosphatases) (14). Changes to the phosphorylation status can induce a rapid change in protein activity, making changes in phosphorylation is an important feature of the initial stages of cell stress responses. In this study, we decided to assess the phosphorylation pattern of a selection of Cdc48 cofactor proteins in yeast. We discovered that one cofactor, Shp1, undergoes robust changes in phosphorylation in response to cell stress. We then went on to perform a detailed

MRC Protein Phosphorylation and Ubiquitylation Unit, School of Life Sciences, University of Dundee, Dundee, UK

Correspondence: arousseau@dundee.ac.uk
Alexander Agrotis's present address is Department of Cell and Developmental Biology, Division of Biosciences, University College London, London, UK

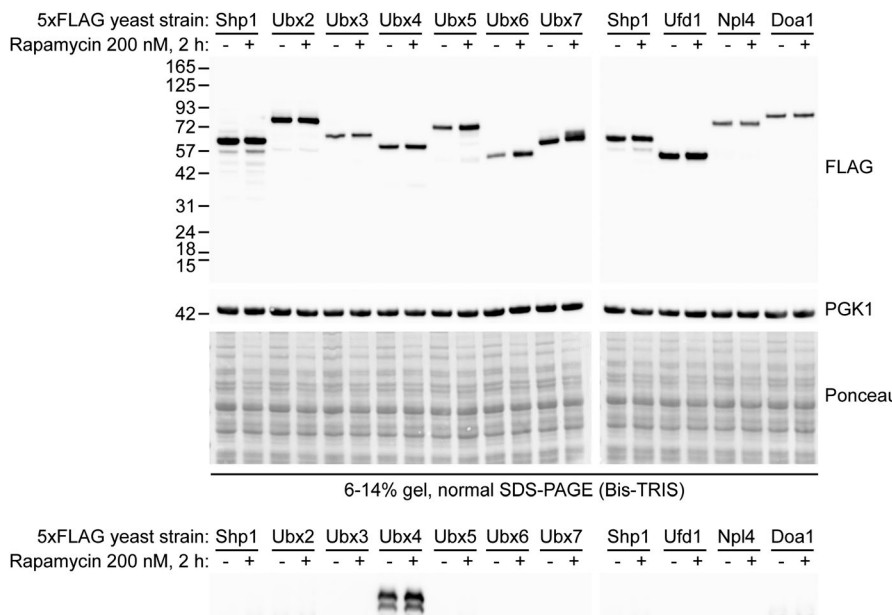

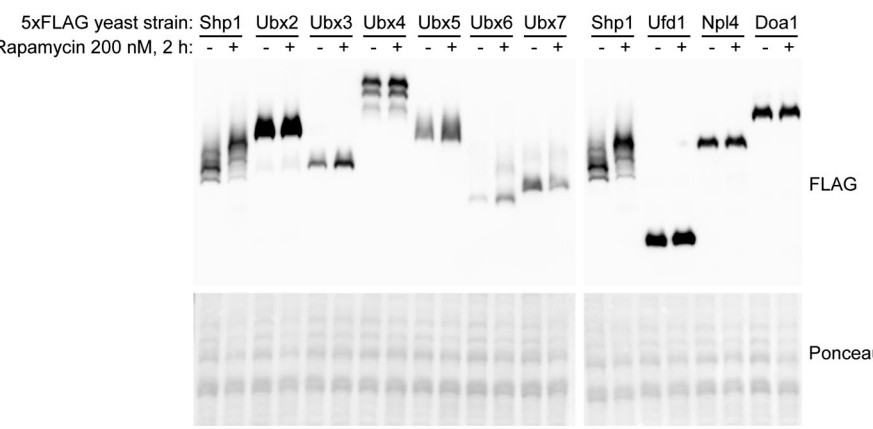

6-14% gel, normal SDS-PAGE (Bis-TRIS)

6.0% gel, 50 μM Phos-tag (Mn²⁺)

**Figure 1.  Phos-tag analysis of endogenous yeast Cdc48 adaptors and Ubx proteins reveals rapamycin-stimulated phosphorylation of Shp1/Ubx1.**
Normal Western blot (upper panel) and Phos-tag Western blot (lower panel) analysis of endogenous 5xFLAG-tagged Ubx and Cdc48 adaptor yeast strains after rapamycin treatment. PGK1 and Ponceau staining were used as loading controls.

characterisation of the regulation and function of this phosphorylation change.

## Results

### Phos-tag analysis of yeast Cdc48 cofactors reveals dynamic phosphorylation of Shp1 in response to TORC1 inhibition

To study the phosphorylation of Cdc48 cofactors including Ubx proteins, we generated *Saccharomyces cerevisiae* strains with a 5xFLAG tag knocked-in to the endogenous coding sequence of key Cdc48 cofactors including Shp1, Ubx2-7, Ufd1, Npl4, and Doa1 (15). This strategy allowed specific detection of endogenous levels of these proteins by Western blotting using anti-FLAG antibody. Because we were interested in identifying Cdc48 cofactors that respond to cell stress, and stresses frequently cause TORC1 inhibition, we cultured these strains in the presence and absence of rapamycin, a TORC1 inhibitor known to induce proteasome assembly and activity (16). The cells were then subjected to lysis under denaturing conditions, and protein phosphorylation patterns assessed by immunoblotting of the same protein extracts run in the presence or absence of Phos-tag (17).

As seen in Fig 1, a subset of the cofactor proteins tested appeared as a single predominant band in both normal and Phos-tag Western blotting, regardless of treatment condition (Ubx2, Ubx3, Ufd1, Npl4, and Doa1), suggesting their phosphorylation status was not substantially altered by rapamycin treatment. We observed a smeared appearance by Phos-tag for Ubx5, Ubx6, and Ubx7 with subtle changes upon rapamycin treatment, suggesting the phosphorylation of these proteins might be slightly affected by rapamycin. Another protein, Ubx4 appeared as multiple bands by Phos-tag, regardless of treatment. Site-directed mutagenesis suggested that none of the previously reported phosphorylation sites for Ubx4 (18, 19, 20) were responsible for this band pattern (Fig S1); therefore, Ubx4 is likely phosphorylated at multiple sites other than those tested (S104, S264, S344, T398, and S399). However, overall, the most striking result was for Shp1 protein (Fig 1), which showed the appearance of multiple bands by Phos-tag that underwent an apparent upward mobility shift after rapamycin treatment. This indicates that rapamycin strongly affects the phosphorylation status of Shp1.

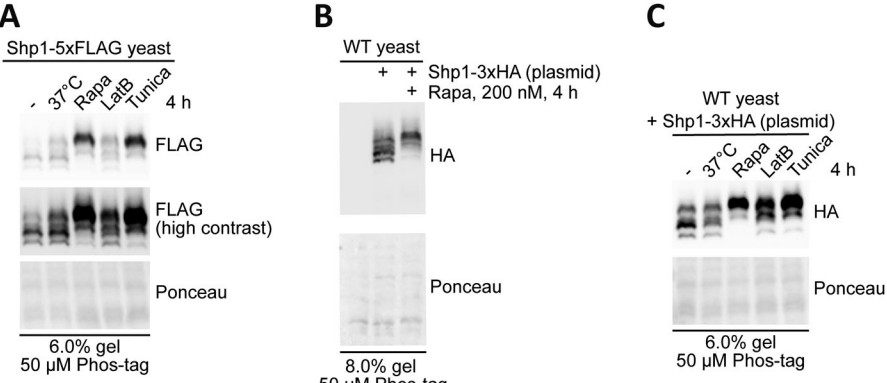

**Figure 2. Various stresses induce the phosphorylation of endogenous-tagged and vector-encoded overexpressed Shp1.**
**(A)** Assessment of endogenous Shp1-5xFLAG phosphorylation by Phos-tag Western blot in yeast after 4 h incubation at 37°C or treatment with 200 nM rapamycin, 5 µM tunicamycin, or 10 µM latrunculin B. **(B)** Validation of phosphorylation of vector-encoded 3xHA-tagged Shp1 by Phos-tag Western blot in yeast after rapamycin treatment. **(C)** Assessment of phosphorylation of vector-encoded 3xHA-tagged Shp1 by Phos-tag Western blot in yeast after 4 h incubation at 37°C or treatment with 200 nM rapamycin, 5 µM tunicamycin, or 10 µM latrunculin B. **(A, B, C)** Ponceau staining was used as a loading control.

## Shp1 phosphorylation increases at both S108 and S315 after TORC1 inhibition

We decided to focus on studying the apparent phosphorylation of Shp1 because this protein showed the most robust and distinctive changes after rapamycin treatment compared with other Cdc48 cofactor proteins. Interestingly, other stresses including ER stress induced by tunicamycin, heat stress and cytoskeletal stress induced by latrunculin B also resulted in elevated Shp1 phosphorylation (Fig 2A), albeit to varying degrees of efficiency compared with rapamycin.

To enable identification of the potential phosphorylated residues on Shp1 protein, we first cloned the Shp1 coding sequence into a vector with a GPD promoter and a C-terminal 3xHA tag. Ectopically expressed Shp1-3xHA in WT yeast showed a similar phosphorylation pattern in response to rapamycin (Fig 2B) when compared with endogenous Shp1-5xFLAG (Fig 2A), indicating that ectopic expression does not alter phosphorylation of the protein. We also observed phosphorylation of Shp1-3xHA in response to other stresses (Fig 2C). We, therefore, deemed vector-encoded Shp1-3xHA a suitable and convenient tool to study potential Shp1 phosphorylation sites.

Because serine–proline (SP)/threonine–proline (TP) motifs are among those reported to be frequently phosphorylated by kinases acting downstream of TORC1 (21), we initially focused on testing the five residues corresponding to SP and TP motifs found in Shp1 (S108, S315, S321, S322, and T331) as they have been previously identified by mass spectrometry as being phosphorylated (18, 20, 22, 23). We generated mutant forms of Shp1-3xHA by changing candidate serine or threonine residues to alanine. Mutation of Shp1 residues S108 and S315 to alanine resulted in a marked downward mobility shift of the uppermost band compared with WT Shp1, when assessed by Phos-tag Western blotting (Fig 3A). This observation was apparent under both untreated and rapamycin-treated conditions, with an increase in intensity of the upper band after rapamycin treatment. In contrast, all other sites tested did not affect Shp1 mobility. Combining S108A and S315A mutations of Shp1-3xHA led to a complete abolition of the protein mobility shift observed by Phos-tag Western blotting, with the appearance of a single predominant band migrating like the lower band of WT Shp1-3xHA in both untreated and rapamycin-treated conditions (Fig 3B).

Mutation of the additional candidate sites had no further additive effect on Shp-3xHA mobility, as shown by the "5A" mutant. It is worth noting that the band corresponding to phosphorylated Shp1 upon rapamycin treatment migrated consistently higher than the uppermost band of Shp1 at steady state. This additional shift is not present in the Shp1-S108A/S315A mutant, indicating that either S108 and/or S315 may act as a priming site for an additional phosphorylation upon stress. Moreover, Shp1-S315A, but not Shp1-S108A, migrated higher upon rapamycin treatment, indicating that S108 is likely responsible for the additional phosphorylation (Fig 3B). Interestingly, peptide harbouring S108 has been shown to be dually phosphorylated at S106 and S108 by mass spectrometry (18, 20, 22, 23), suggesting that S108 could prime the phosphorylation at S106. By analysing the impact of S106 mutation, we observed that the higher migrating band was lost in Shp1-S106A upon rapamycin treatment compared with its WT counterpart (Fig S2). Moreover, S106A mutation prevented rapamycin-induced phosphorylation of Shp1-S315A (Figs 3B and S2). These results show that S108 is likely a priming site for S106. Overall, our data indicate that yeast Shp1 is phosphorylated at either of the two residues S108 and S315 under non-stressed conditions but, following stress, most of the Shp1 becomes phosphorylated at both residues with S108A being important for the additional phosphorylation of S106.

Although it is likely S108 and S315 are phosphorylated, the band shift observed could be due to an alternate modification. To confirm that the Phos-tag mobility shift of Shp1 was because of phosphorylation, we treated lysates from rapamycin-treated Shp1-5xFLAG yeast with Λ protein phosphatase in the presence and absence of the phosphatase inhibitor cocktail PhosSTOP (Fig 3C). This resulted in a progressive downward mobility shift of Shp1 until it formed a single predominant band. This shift was slowed by the presence of PhosSTOP, confirming that dephosphorylation was being observed. The lower band was identical in mobility to endogenous Shp1-5xFLAG with S108A and S315A mutations engineered using CRISPR/Cas9 (double mutant of S108A and S315A, hereafter abbreviated as "2SA") (Fig 3C). Therefore, the mobility pattern of Shp1 observed by Phos-tag Western blotting is indeed due to differential phosphorylation at residues S106, S108, and S315, residues which are all located in interdomain regions of the protein (Fig 3D) (5). The total levels of WT and 2SA Shp1-5xFlag remained

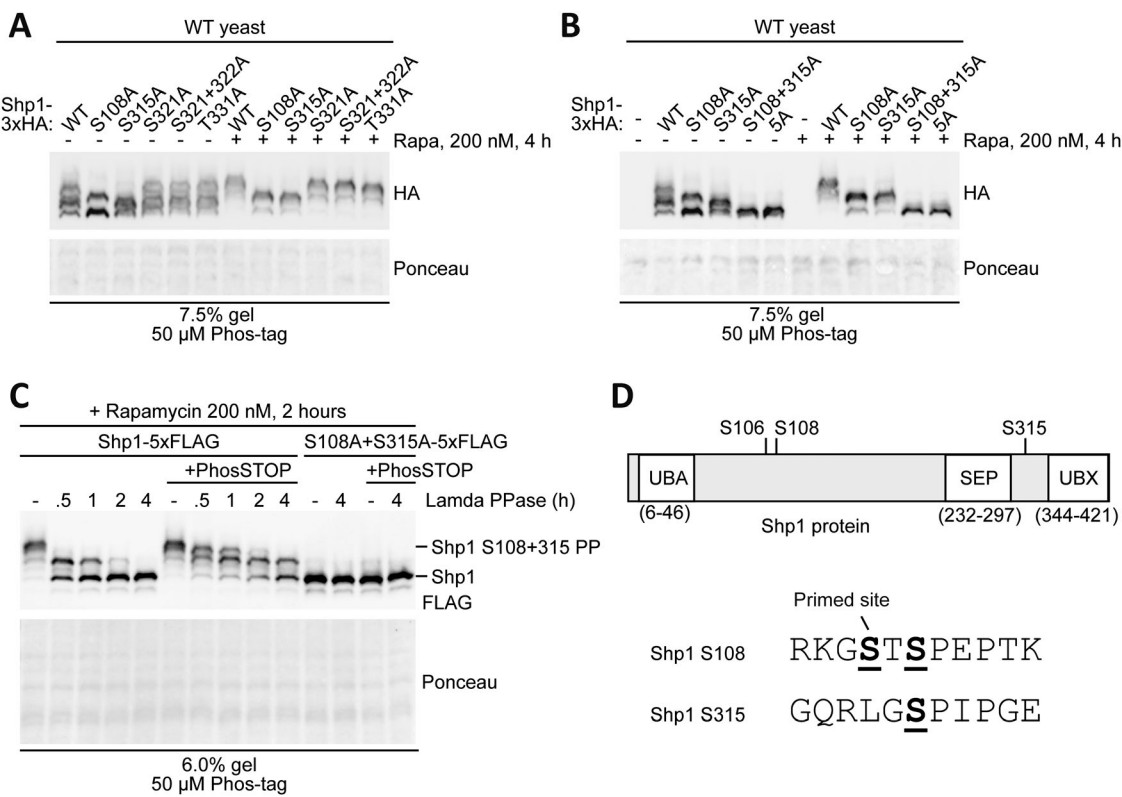

**Figure 3. Site-directed mutagenesis and phosphatase treatment confirm that Shp1 is phosphorylated at both S108 and S315 after stress.**
**(A)** Phos-tag analysis of yeast expressing single phosphomutant forms of vector-encoded Shp1-3xHA, with and without rapamycin treatment. **(B)** Phos-tag analysis of yeast expressing single (S108A, S315A), double (S108 + S315A), and 5A (S108A + S315A + S321A + S322A + T331A) phosphomutant forms of vector-encoded Shp1-3xHA, with and without rapamycin treatment. **(C)** Phosphatase treatment of non-denaturing lysates from rapamycin-treated Shp1-5xFLAG and Shp1 S108A + S315A-5xFLAG (CRISPR/Cas9 edited) yeast. The phosphorylation status was assessed by Phos-tag Western blotting. **(A, B, C)** Ponceau staining was used as a loading control. **(D)** Protein domain map (not to scale) adapted from reference 5 and surrounding peptide sequence of phosphorylation sites on Shp1.

unchanged under both unstressed and stressed conditions, indicating that Shp1 phosphorylation does not regulate the Shp1 level (Fig S3A). Moreover, Shp1 phosphomutants only showed slight defects in rescuing the growth of shp1Δ cells upon rapamycin treatment, indicating that the function of Shp1 is not completely abolished by the loss of phosphorylation (Fig S3B).

### Shp1 phosphorylation status does not affect binding to Cdc48 or autophagy

We considered the possibility that phosphorylation could be a way of regulating the binding of Shp1 to Cdc48. To test this hypothesis, we engineered a yeast strain with a 5xFLAG tag at the C-terminus of endogenous Cdc48 to enable specific detection using a FLAG antibody, before transforming this strain with vectors encoding Shp1-3xHA WT or 2SA phosphomutant (Fig 4A). This allowed us to use immunoprecipitation (IP) to assess the level of Cdc48-5xFLAG binding to Shp1-3xHA. During extraction of Shp1-3xHA under non-denaturing conditions we found that Shp1 is highly sensitive to proteases, but inclusion of both protease inhibitor cocktail and PMSF reduced its sensitivity to proteolytic cleavage (Fig S3C).

We performed IP of Shp1-3xHA and 2SA phosphomutant before probing for Cdc48-5xFLAG. We could detect a specific interaction versus cells not expressing Shp1-3xHA (Fig 4B); however, the level of Cdc48 pulldown was indistinguishable between Shp1-3xHA WT and 2SA. This was also the case when the experiment was carried out in the presence of rapamycin to stimulate Shp1-3xHA WT phosphorylation (Fig 4C). To eliminate the possibility that dephosphorylation occurred during purification, we used Phos-tag to confirm that Shp1-3xHA remained phosphorylated after IP (Fig 4D). Therefore, our data suggest that Shp1 phosphorylation at S108 and S315 does not play an important role in regulating the binding between Shp1 and Cdc48.

Because Shp1 has been implicated in autophagy (12), we also tested the role of Shp1 phosphorylation in autophagy using a GFP-Atg8 processing assay (24). In this assay, free GFP accumulation occurs after autophagy as a result of GFP-Atg8 breakdown in the vacuole. After 2 h of rapamycin treatment we detected a clear increase in free GFP accumulation in WT yeast but not autophagy-defective atg1Δ yeast, as expected (Fig 4E and F). Autophagy is also defective in shp1Δ yeast, as previously described (12). Endogenous Shp1 2SA phosphomutant strain did not display any noticeable defect in autophagy, suggesting that Shp1 phosphorylation is not required for autophagy. Overall, we find that Shp1 phosphorylation at S108 and S315 is dispensable for Cdc48 binding and autophagy.

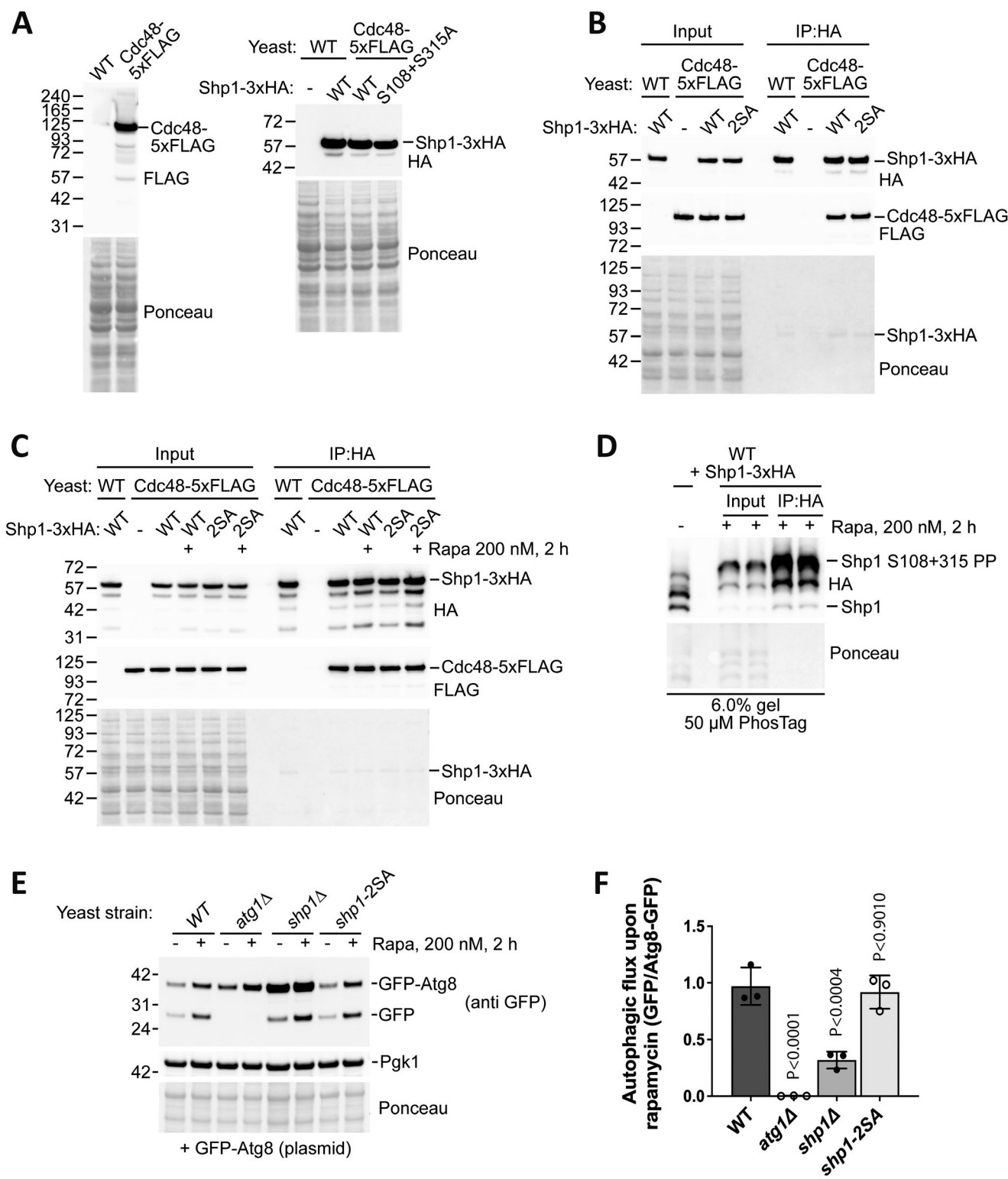

**Figure 4. Loss of Shp1 phosphorylation does not affect Cdc48 binding or autophagy.**
**(A)** Western blot validation of Cdc48-5xFLAG strain generation (left panel) and expression level of vector-encoded Shp1-3xHA WT and 2SA (S108A + S315A) phosphomutant (right panel). **(B)** IP of Shp1-3xHA WT and 2SA phosphomutant expressed in WT and Cdc48-5xFLAG yeast, using anti-HA beads. Specific pulldown of Cdc48-5xFLAG in the presence of Shp1-3xHA was detected by Western blotting. **(B, C)** IP experiment as described in (B), but with the addition of samples treated for 2 h with rapamycin to assess the binding of Shp1-3xHA to Cdc48-5xFLAG after stress-induced Shp1 phosphorylation. **(D)** Phos-tag analysis of Shp1-3xHA in rapamycin-treated cells before and after anti-HA immunoprecipitation. An untreated sample using the standard denaturing lysis method was included as a reference to verify the identity of the unphosphorylated and double-phosphorylated Shp1 bands. **(E)** GFP-Atg8 processing assay after 2 h rapamycin treatment to induce autophagy (which induces free GFP

### Abolishing Shp1 phosphorylation does not affect the role of Shp1 in the cadmium stress response

Shp1 is known to play an important role in response to cadmium toxicity by working with Cdc48 to promote the disassembly of the E3 ligase complex SCF$^{Met30}$ (5). Interestingly, we found that treating yeast with cadmium robustly increased Shp1 phosphorylation similar to rapamycin treatment (Fig 5A). Therefore, we next wanted to test whether Shp1 phosphorylation is important for influencing the activity of SCF$^{Met30}$ upon cadmium stress. To monitor SCF$^{Met30}$ activity, we generated a collection of yeast strains with endogenous Met4 (a ubiquitination substrate and downstream effector of SCF$^{Met30}$) tagged with 5xFLAG. As described previously, cadmium treatment triggers rapid Met4 deubiquitination due to the loss of ubiquitination by SCF$^{Met30}$ after Shp1-mediated disassembly (5). Therefore, deubiquitination of Met4 upon cadmium treatment is indicative of Shp1 function in this process. As seen in Fig 5B, 20 min of cadmium treatment resulted in a marked reduction in ubiquitinated Met4-5xFLAG in WT yeast, while this reduction was greatly impaired in shp1Δ yeast as expected. However, Shp1 2SA phosphomutant yeast had comparable SCF$^{met30}$ activity to WT yeast. We also assessed the rescue of the shp1Δ Met4-5xFLAG strain and found that rescue with either WT or 2SA Shp1 expressed under its own promoter was sufficient to restore Met4 deubiquitination in response to cadmium (Fig 5C). Moreover, after 2 h treatment with cadmium, difference in Met4 ubiquitination was not observed in shp1Δ cells compared with their WT counterpart, indicating that Shp1 loss slows the deubiquitination kinetics of Met4, rather than abolishing it, upon cadmium stress (Fig 5D). These results, combined with our observation that Shp1 phosphorylation does not appear to increase within 20 min of cadmium stress (Fig 5E), suggest that Shp1 phosphorylation at S108 and S315 does not play an important role in SCF$^{Met30}$ disassembly.

### Mpk1 kinase and the PP1 phosphatase catalytic subunit Glc7 are regulators of the Shp1 phosphorylation level

We next decided to identify mediators of Shp1 phosphorylation at S108 and S315 to determine pathways involving Shp1 phospho-regulation. Shp1-3xHA was expressed in yeast strains lacking different non-essential kinases and phosphatases, and Shp1 phosphorylation was monitored by Phos-tag Western blotting. We focused on testing MAP kinases, as they often phosphorylate SP/TP motifs (25) which correspond to those identified in Shp1 (see Fig 3D) and kinases and phosphatases linked to TORC1 function (21) because Shp1 phosphorylation is enhanced by rapamycin treatment (Fig S4).

Yeast lacking Atg1, Ypk1-3, Rim15, Sit4, Gcn2, Pkh1-2, Tip41, Yak1, or Sky1 showed no impairment in Shp1 phosphorylation (Fig S4). Also, most of the MAP kinases tested (including Hog1, Fus3, Kss1, and Smk1) had no noticeable effect on basal or rapamycin-induced

Shp1 phosphorylation when knocked out. However, loss of the MAP kinase Mpk1/Slt2 reduced the ratio of double-phosphorylated Shp1 to unphosphorylated Shp1 after rapamycin treatment, compared with WT yeast (Fig S4). This was confirmed by monitoring the status of Shp1 phosphorylation over time after rapamycin treatment (Fig 6A). We tested single phosphomutant forms of Shp1-3xHA to compare how Mpk1 affects S108 or S315 phosphorylation, but there was no clear effect on either site by Mpk1 loss (Fig 6B). We also generated mpk1Δ cells with endogenous-tagged Shp1-5xFLAG to determine how Mpk1 affects the phosphorylation of Shp1 at endogenous levels. We observed that rescue of mpk1Δ yeast with Mpk1-6xHis restored normal levels of Shp1 phosphorylation after 3 h of rapamycin treatment, compared with rescue with empty vector or catalytic-inactive Mpk1 K54R (Fig 6C). Therefore, Mpk1 positively regulates the phosphorylation of Shp1 upon rapamycin treatment, and this regulation requires the catalytic activity of Mpk1.

Increased Shp1 phosphorylation after stress might also involve reduced dephosphorylation. Shp1 is known to interact with and regulate quality control of the essential PP1 phosphatase subunit Glc7 (11, 13), making this a candidate for Shp1 dephosphorylation. We monitored Shp1-3xHA phosphorylation in the temperature-sensitive Glc7 strain glc7-12 (26) and found that growth at the non-permissive temperature was sufficient to induce Shp1 phosphorylation at considerably higher levels than WT cells (Fig 7A). Analysis of single site Shp1 phosphomutants revealed that both S108 and S315 showed increased phosphorylation after Glc7 inactivation (Fig 7B). Consistently, Glc7 inactivation further caused increased phosphorylation of endogenously tagged Shp1-5xFLAG (Fig 7C). This could be prevented by re-expressing WT Glc7-3xHA (Fig 7D). Moreover, Shp1 abundance remained unaffected by inhibition of Glc7, indicating that the regulation mainly occurs at the phosphorylation level (Fig 7E). Therefore, Glc7 is a likely regulator of Shp1 phosphorylation. Whether Glc7 directly dephosphorylates Shp1, or its loss induces a stress that indirectly triggers Shp1 phosphorylation, remains to be determined.

Overall, this survey of the phosphorylation status of multiple Cdc48/p97 cofactors identified two phosphorylation sites on yeast Shp1 (S108 and S315), which markedly increase in the phosphorylation level upon various cell stresses. Although these phosphorylation sites seem to be dispensable for known functions of Shp1, we have identified two regulators of Shp1 phosphorylation: the MAP kinase Mpk1 and PP1 phosphatase catalytic subunit Glc7. Knowledge of these regulators could potentially lead to the discovery of novel Shp1 functions.

## Discussion

While numerous post-translational modification sites of p97/Cdc48 and its cofactors have been reported since the emergence of large-scale proteomics datasets, most of these sites have not been

accumulation as a marker of autophagy). GFP-Atg8 plasmid was transformed in yeast strains including WT, shp1Δ strain, and Shp1 2SA phosphomutant strain (S108A, S315A, and PAM mutation), and autophagy-defective atg1Δ strain as a negative control. **(A, B, C, D, E)** Ponceau staining was used as a loading control. **(E, F)** Quantification of the autophagy level (GFP/Atg8-GFP) from (E) after rapamycin treatment. n = 3 independent biological replicates. Data are shown as mean ± SD. P-values were calculated using one-way ANOVA test.

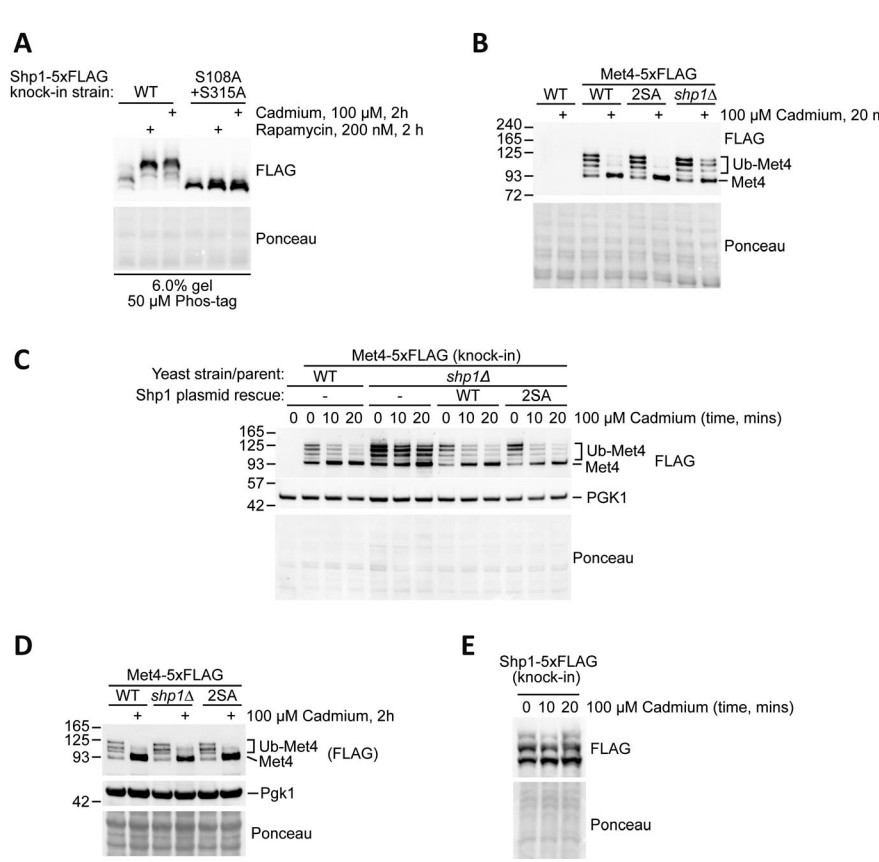

**Figure 5. Shp1 phosphorylation occurs in response to cadmium stress but does not affect the deubiquitination of Met4.**
**(A)** Assessment of endogenous Shp1-5xFLAG phosphorylation in yeast after rapamycin and cadmium treatment by Phos-tag Western blotting. **(B)** Assessment of endogenous 5xFLAG-tagged Met4 deubiquitination after cadmium treatment in the background of *shp1Δ* and Shp1 phosphomutant (2SA). Met4-5xFLAG was detected by Western blotting using anti-FLAG antibody. **(C)** Assessment of endogenous 5xFLAG-tagged Met4 deubiquitination after cadmium treatment in WT and *shp1Δ* (NatMX) strain background after plasmid rescue with an empty vector (p416 GPD, –), or Shp1 with its own promoter and UTRs (WT and 2SA phosphomutant version). Met4-5xFLAG was detected by Western blotting using anti-FLAG antibody. PGK1 was used as a loading control. **(A, B, C)** Ponceau staining was used as a loading control. **(D)** Assessment of endogenous 5xFLAG-tagged Met4 deubiquitination after 2 h cadmium treatment in the background of *shp1Δ* and Shp1 phosphomutant (2SA). Met4-5xFLAG was detected by Western blotting using anti-FLAG antibody. **(E)** Assessment of endogenous Shp1-5xFLAG phosphorylation in yeast after cadmium treatment by Phos-tag Western blotting.

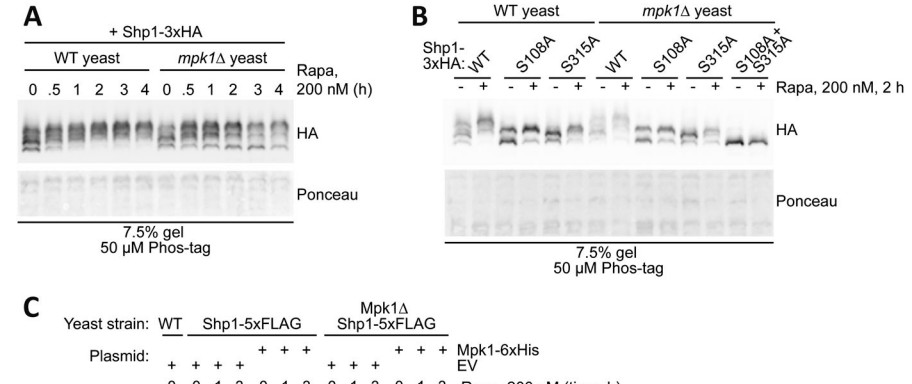

**Figure 6. Mpk1 kinase influences Shp1 phosphorylation at S108 and S315.**
**(A)** Rapamycin treatment time course of Shp1-3xHA phosphorylation in WT and Mpk1 knockout yeast, assessed by Phos-tag. **(B)** Phos-tag assessment of vector-encoded Shp1-3xHA single and double phosphomutant phosphorylation in WT and *mpk1Δ* cells after rapamycin treatment. **(C)** Phos-tag assessment of phosphorylation of endogenous tagged Shp1-5xFLAG in WT versus *mpk1Δ* background. Strains were transformed with plasmids p416 GPD (empty vector) and p416 GPD Mpk1-6xHis and treated with rapamycin for the indicated time before lysis. **(A, B, C)** Ponceau staining was used as a loading control.

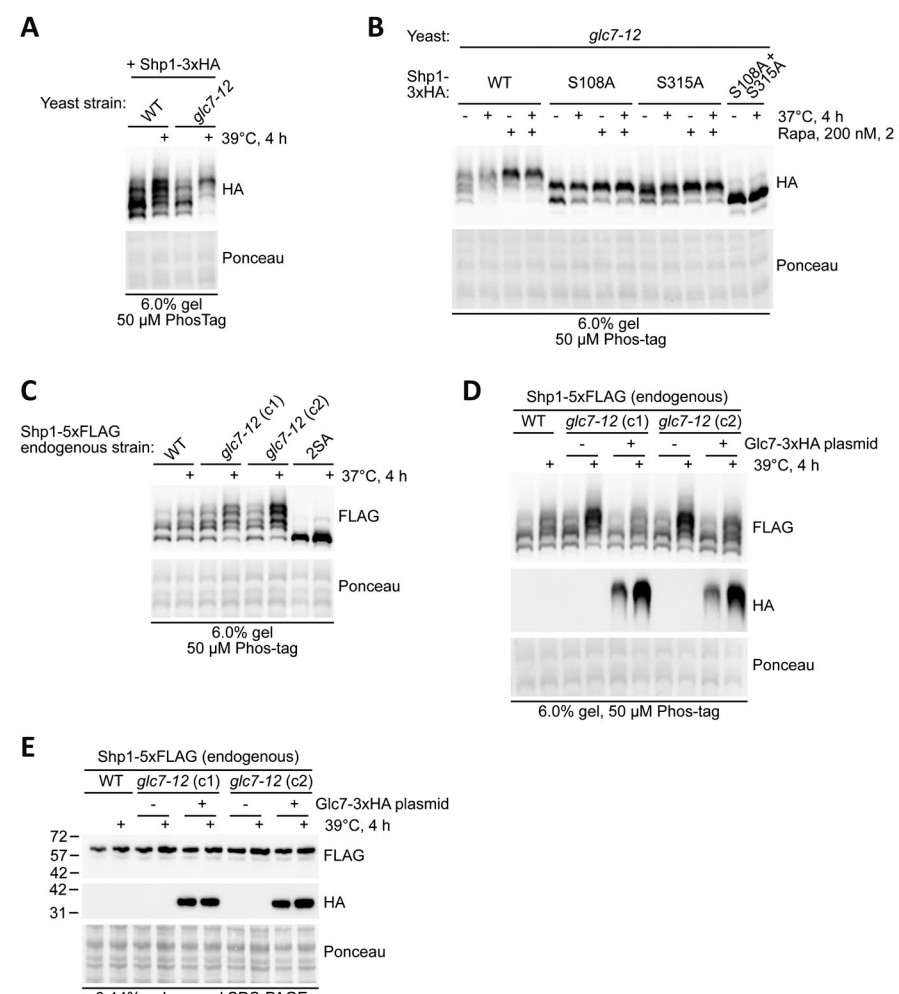

**Figure 7. PP1 phosphatase catalytic subunit Glc7 regulates the phosphorylation of Shp1 at sites S108 and S315.**

**(A)** Phos-tag analysis of vector-encoded Shp1-3xHA phosphorylation in WT versus temperature-sensitive *glc7-12* mutant yeast strain. Strains were cultured at 24°C after 4 h growth at a non-permissive temperature (39°C). **(B)** Phos-tag analysis of vector-encoded Shp1-3xHA WT, single and double phosphomutants overexpressed in *glc7-12* yeast grown at a non-permissive temperature (37°C) for 4 h, with and without rapamycin treatment for the final 2 h before lysis. **(C)** Assessment of endogenous Shp1-5xFLAG phosphorylation in WT versus *glc7-12* genetic background after 4 h growth at a non-permissive temperature (37°C). The samples were analysed by Phos-tag Western blotting. c1: clone 1 and c2: clone 2. **(D)** Assessment of endogenous Shp1-5xFLAG phosphorylation in WT and *glc7-12* genetic background expressing either the empty vector or a vector encoding WT Glc7-3xHA after 4 h growth at a non-permissive temperature (37°C). The samples were analysed by Phos-tag Western blotting. c1: clone 1 and c2: clone 2. **(D, E)** Western blot analysis of Shp1-5xFLAG and Glc7-3xHA from sample analysed in (D). **(A, B, C, D, E)** Ponceau staining was used as a loading control. c1: clone 1 and c2: clone 2.

studied in detail (27). Because these proteins play important roles in cell stress resistance (28), it is of great interest to understand how they are regulated, with phosphorylation being a key, and widespread, stress-responsive protein regulatory mechanism.

By carrying out Phos-tag analysis on yeast Cdc48 cofactor proteins, we were able to readily detect qualitative changes in their phosphorylation pattern upon cell stress. This highlighted Shp1 as being unique among the Cdc48 cofactors tested, with clear stress-induced phosphorylation. Strikingly, despite there being over 25 Shp1 phosphorylation sites reported on Saccharomyces Genome Database (http://www.yeastgenome.org) to date, phosphorylation at residues S108 and S315 account for almost the entire pool of phosphorylated Shp1 under normal and stress conditions, with the exception of S106 site, which is selectively phosphorylated upon stress.

We identified phosphorylations from other Cdc48 cofactors, which are worthy of further study. Ubx4, known to affect Cdc48 activity and function in ERAD, appears to be heavily phosphorylated (29). Identifying the phosphorylation sites, their functions, and regulation will be important to gain further insights into the cellular role of Ubx4. Notably, the phosphorylation sites are also distinct

from those identified by phosphoproteomics data. Furthermore, Ubx5, Ubx6, and Ubx7 each exhibited subtle shifts in phosphorylation, which may be functionally important.

Although we have identified Mpk1 as a regulator of Shp1 phosphorylation at S108 and S315, Mpk1 loss only partially reduces phosphorylation. This could be due to either redundancy with other stress-activated kinases, or compensation by those self-same kinases in the *mpk1Δ* strains used. Under either scenario, multiple kinases may need to be knocked out to observe a complete loss of Shp1 phosphorylation. Both S108 and S315 phosphosites harbour the following SPxP (S: serine, P: proline, and x: any residue) motif. This motif has been previously identified using an algorithm designed to extract motifs from large data sets of naturally occurring phosphorylation sites in humans. Although this SPxP motif is known to be phosphorylated in proteins (e.g., serine 163 of lymphocyte phosphatase-associated phosphoprotein), the kinase(s) responsible for its phosphorylation remain(s) unknown (30, 31). It has been suggested that proline-directed kinases such as CDKs and MAP kinases could recognise and phosphorylate this motif (30). In agreement with this, we have shown that Mpk1 is important for Shp1 phosphorylation at these two SPxP sites,

although whether this is direct or indirect remains to be elucidated. Indeed, one could imagine that Mpk1 phosphorylates and hence activates another kinase phosphorylating Shp1 upon stress. Moreover, Mpk1 loss mainly leads to an accumulation of the non-phosphorylated form of Shp1. This could indicate that Mpk1 is important for the initial phosphorylation of Shp1, whereas other kinases perform subsequent addition of other phosphates. We have also tested four other MAP kinases (Hog1, Fus3, Kss1, and Smk1) but none of them seemed to be involved in Shp1 phosphorylation. Yeasts have six CDKs (Cdc28, Pho85, Kin28, Srb10, Bur1, and Ctk1) and it will be important to further study their potential involvement in Shp1 phosphorylation.

Mpk1 belongs to the cell wall integrity signalling pathway, which is responsible for the maintenance and biosynthesis of the cell wall and has been linked to various other cellular processes such as cell division, autophagy, proteasome assembly, and plasma membrane homeostasis (32, 33). As Mpk1 contributes to stress-induced Shp1 phosphorylation, Shp1 could play a role downstream of this kinase. Mpk1 and Shp1 regulate common important processes (e.g., protein degradation by the proteasome, stress resistance, and autophagy), making it possible that Shp1 is an effector protein of the Mpk1 signalling pathway. In agreement with this hypothesis, negative genetic interactions have been reported between these two genes, suggesting that they are epistatic (34, 35). Further study will help understanding the crosstalk between Mpk1 and Shp1 upon cell stress.

In addition to a regulatory role for Mpk1, we identified Glc7 as being functionally important for the Shp1 phosphorylation status. Whether Glc7 functions are upstream, downstream, or independent of Mpk1 is yet to be determined. However, crosstalk between Mpk1 and Glc7 has been reported, with the overexpression of Pck1, an upstream activating kinase of Mpk1, suppressing the growth defect of the glc7-10 temperature-sensitive mutant at the non-permissive temperature. Moreover, the deletion of MPK1 gene is synthetic lethal in the glc7-10 temperature-sensitive mutant (26, 36). Collectively, this indicates that Mpk1 and Glc7 operate in shared signalling pathways, including Shp1 phosphorylation. Although no mechanism has been reported, we could imagine that either Mpk1 or Glc7 has an antagonistic function by regulating the phosphorylation status of the same subset of substrates, or that one protein is acting upstream of the other regulating its activity.

Our discovery that Glc7 inactivation enhances Shp1 phosphorylation raises an intriguing possibility: that stress conditions might promote Shp1 phosphorylation through reduced dephosphorylation. Shp1 has been shown in several studies to regulate PP1 complex assembly and thus Glc7 function (11, 13, 37). Moreover, the deletion of Shp1 and Glc7 is synthetic lethal, whereas a mutant of Shp1 unable to bind Cdc48/p97 supresses the toxicity associated with Glc7 overexpression (11, 38). Together, these studies suggest that Shp1 mediates the recruitment of Glc7 to Cdc48/p97, a pre-requisite for Glc7 activation. In agreement with this, human Cdc48/p97 functions with the Shp1 homologs (p37, p47, and UBXN2A) to dissociate the I3 maturation factor from immature Gcl7 complexes (39). We now present the first evidence suggesting that the reverse might also be true (that Glc7 activity regulates Shp1 phosphorylation). We could imagine a scenario in which phosphorylated Shp1 recruits immature Glc7 complex to Cdc48/p97 to remove maturation

factors and, thereby, activate Glc7. Active Glc7 would in turn dephosphorylate Shp1 promoting its release from Cdc48/p97. An alternative indirect scenario could be that Glc7 loss triggers a stress, activating a kinase-phosphorylating Shp1. Further studies will help in understanding the reciprocal regulation between Shp1 and Glc7 complex.

In this study we were not able to identify a function for Shp1 phosphorylation as S108 and S315. There are several possible reasons for this. Shp1 phosphorylation could play a role in an undiscovered function of Shp1 that we have not tested. Alternatively, if loss of Shp1 phosphorylation is particularly deleterious, compensatory mechanisms could have been acquired, as has been observed for other mutants (40). Finally, there is the possibility that Shp1 phosphorylation, at these sites, has no significant impact on Shp1 function. Better understanding the role of Shp1 in response to stress will be key to answer this question.

In conclusion, in this study, we have streamlined a system of endogenous tagging and Phos-tag Western blotting to assess the phosphorylation status of multiple Cdc48 cofactor proteins. We identified Shp1 as being differentially phosphorylated upon stresses including TORC1 inhibition, ER stress, and cadmium toxicity. We determined the sites as S108 and S315, with double phosphorylation becoming predominant under the aforementioned cell stresses. Shp1 phosphoregulation is temporally regulated upon cell stress by Mpk1 kinase and Gcl7 phosphatase.

## Materials and Methods

### Yeast strain generation and CRISPR modification

All *S. Cerevisiae* strains in this study used the S288C-based parent strain BY4741 (ATCC number 201388) referred to as "WT." Deletion strains using the KanMX marker were obtained from the Thermo Fisher Scientific (Mat-A-Set) collection (cat: 95401.H2), except the *shp1Δ* strain in Fig 5C which was generated for this study using the NatMX resistance cassette. Temperature-sensitive *glc7-12* strain was described previously (26). Endogenous 5xFLAG-tagged knock-in strains were generated by amplification of the 5xFLAG-His3MX cassette from pKL259, a gift from Karim Labib's laboratory. All primers used for strain generation are provided in Supplemental Data 1. Yeast was transformed using a lithium acetate/PEG/carrier DNA transformation protocol (41) with heat shock at 42°C for 15 min.

CRISPR-modified yeast were generated using a system described previously (42). Guide RNA was selected using Wyrick's lab yeast gRNA search engine (Washington State University) to cut next to the codon for Shp1 G209 (fw: gatcctggaagaggttttagattagtttttagagctag, rv: ctagctctaaaactaatctaaaacctcttccag) and inserted into pML104 (Addgene number: 67638) to generate pML104-Shp1. The repair template was amplified from a vector encoding Shp1 harbouring S108A and S315A mutations and where the PAM motif had been mutated (without affecting the amino acid sequence) to prevent further cleavage by Cas9 using the following primers: f: caagaagcttctccaccaac and r: cagtgttcgcatttgatgtcac. pML104-Shp1 (500 ng) was co-transformed in yeast with 50 µl of repair template PCR

reaction and plated on URA plates. The clones were verified by PCR using flanking primers (fw: caagaagccttctccaccaac, rv: cagtgttcg-catttgatgtcac) and confirmed by sequencing, then grown on 5-FOA plates to select against the expression of Cas9 before experiments. The CRISPR Shp1 phosphomutant (2SA) strain was subsequently endogenously tagged with 5xFLAG, as described above to generate S108A + S315A-5xFLAG.

### Plasmid cloning and DNA purification

All plasmid cloning and mutagenesis were performed by PCR and InFusion recombination. GFP, Atg8 CDS, and the 5′ and 3′ UTR of Atg8 were amplified in separate fragments and cloned into pRS413 to make pRS413-GFP-ATG8. Shp1 rescue vector plasmid was generated by amplifying the full *SHP1* gene (CDS plus flanking 1,000 nt upstream and 300 nt downstream) and cloning it into p416 without the GPD promoter. All other coding sequences were amplified from WT yeast genomic DNA and cloned into EcoRI-cut p416 GPD, which contains the URA3 marker for selection. For 3xHA C-terminal tagging, this was done by overlap-extension PCR (separate amplification of CDS and 3xHA tag, followed by joining amplification before insertion in p416 GPD cut with EcoRI, using gene-specific 5′ and internal primers and the common 3′ primer "3xHA in GPDEco Rv"). Glc7-3xHA was cloned using a template lacking the Glc7 intron.

PCRs were performed using KOD Hot Start DNA polymerase (71086; Sigma-Aldrich), gel extractions done using Macherey-Nagel Nucleospin Gel and PCR clean up kit (12303368; Thermo Fisher Scientific), and cloning reactions carried out using In-Fusion HD Cloning Plus (638909; Takara), all according to the manufacturers' protocols. Plasmids and cloning products were transformed in Stellar competent cells (636763; Takara), and plasmid DNA was purified using QIAprep Spin MiniPrep Kit (27106; QIAGEN). Constructs were validated by restriction digests and Sanger sequencing (MRC PPU DNA Sequencing and Services). All primers used for cDNA cloning and mutagenesis are provided in Supplemental Data 1.

### Yeast culture and treatments

Yeast was cultured at 30°C unless otherwise stated. Strains were inoculated in 5-ml YEPD (1% yeast extract, 2% peptone, and 2% glucose) or 2% glucose HIS−/URA drop out synthetic medium (for strains expressing plasmids) overnight before experiments. On the day of experiments, yeast was cultured in fresh YEPD (10–30 ml) from a starting density of 0.2 OD for 3–4 h to reach exponential growth phase. Cultures were then readjusted to 0.2 OD and grown for further 2–4 h, with treatments added to media at final concentrations as indicated in figure legends. Experiments were stopped by placing cultures on ice and centrifuging an estimated 2–4 OD of yeast (for Western blotting or Phos-tag) or 8–12 OD (for IP) at 3,200$g$ for 4 min at 4°C. Yeast pellets were washed once in cold water and pelleted at 6,200$g$ for 1 min at 4°C with supernatant removed, before proceeding to lysis (for IP) or snap freezing pellets on dry ice for storage at −20°C (before Western blotting or Phos-tag).

### Antibodies and reagents

Reagents used were as follows: rapamycin (S1039; Selleckchem), tunicamycin (ab120296; Abcam), latrunculin B (ab144291; Abcam), PMSF solution 0.1 M in ethanol (93482; Sigma-Aldrich), N-ethyl-maleimide (NEM) (04259—prepared as 1 M solution in ethanol; Sigma-Aldrich), cadmium chloride hemi(pentahydrate) (239208—prepared as 100 mM stock in water; Sigma-Aldrich).

Antibodies and dilutions used were as follows: HA (1:5,000, 26183; Invitrogen), FLAG M2 (1:2,000, F1804-200UG; Sigma-Aldrich), GFP (1: 200, S268B; MRC PPU Reagents & Services), Mpk1 total (1:500, sc-374434; Santa Cruz), p-Mpk1 (1:1,000, 4370; Cell Signaling Technology), and PGK1 loading control (1:5,000, ab113687; Abcam). Secondary antibodies were anti-mouse/rabbit IgG HRP-linked (7076S/7074S; Cell Signaling Technology) and goat anti-mouse IgG (H + L) secondary antibody DyLight 800 (SA5-35521; Thermo Fisher Scientific).

### Phos-tag Western blotting

On ice, yeast pellets were resuspended in 400 μl 2 M LiAc, pelleted at 6,200$g$ for 1 min, and resuspended in 400 μl 0.4 M NaOH. After pelleting again, yeast were lysed in 120 μl of denaturing Phos-tag lysis buffer (0.1 M NaOH, 2% SDS, and 2% $β$-mercaptoethanol) and boiled at 95°C for 10 min. Acetic acid was added to each sample at a final concentration of 0.1 M before centrifugation at 15,000$g$ for 10 min. The protein concentration of the resulting supernatant was determined using the Protein A280 function of a NanoDrop 1000 (Thermo Fisher Scientific), and the samples were diluted to equal concentrations of 2–5 μg/μl. The samples were mixed with 5× loading buffer (0.25 M Tris–HCl pH 6.8, 50% glycerol, and 0.05% bromophenol blue) and ~30–50 μg of each sample was loaded on a 15 lane 6% acrylamide (unless otherwise stated) gel containing 50 μM Phos-tag acrylamide and 100 μM manganese chloride. Gels were run and washed with and without EDTA before transferring protein onto 0.2 μm nitrocellulose membrane. Full gel preparation, running, and transfer details are provided in Supplemental Data 1. After transfer, membranes were processed as described for normal Western blotting.

### Normal Western blotting (without Phos-tag)

Yeast pellets were lysed and processed as described for Phos-tag Western blotting. The samples were loaded on home-made 6–14% acrylamide gradient Bis–Tris gels (0.33 M Bis–Tris pH 7.5) alongside broad molecular weight (10–245 kD) prestained protein ladder (ab116028; Abcam) and run at 130 V for 110 min using MES-SDS running buffer (Formedium MES-SDS5000). Gels were incubated in transfer buffer (NuPAGE Transfer Buffer with 20% ethanol final, NP0006; Thermo Fisher Scientific) and transferred onto 0.2-μm nitrocellulose membrane (1620112; Bio-Rad) at 25 V constant for 30 min using semidry transfer apparatus (1704150; Bio-Rad). Membranes were stained with Ponceau S solution (sc-301558; Santa Cruz) before imaging on a Bio-Rad ChemiDoc MP System to confirm equal protein loading. Membranes were then washed in Tris-buffered saline (TBS: 50 mM Tris–Cl, pH 7.5, 150 mM NaCl) and blocked in 5% nonfat dry milk in TBS. After further washes in TBS,

membranes were incubated overnight at 4°C in primary antibody (diluted in 4% BSA, 0.02% sodium azide in TBS). After washing in TBS-T (TBS with 0.1% Tween-20), membranes were incubated in secondary antibody (diluted in TBS-T) for 1 h at room temperature, before washing again in TBS-T and developing using Clarity (1705061; Bio-Rad) or Clarity Max (1705062; Bio-Rad) ECL reagent on a Bio-Rad ChemiDoc MP system.

### IP

All steps were performed on ice with chilled reagents. Cell pellets were resuspended in 400 µl IP lysis buffer (100 mM NaCl, 50 mM Tris 7.5, 1% Triton, supplemented with freshly added: 5 mM DTT, 20 mM NEM, 1 mM PMSF, Roche Complete Protease Inhibitor Cocktail EDTA-free and PhosSTOP at recommended concentrations). For FLAG IP, DTT was excluded from the IP lysis buffer. The resuspension was added to an equal volume of glass beads (G8772-500G; Sigma-Aldrich) in 2-ml tubes, and the cells were broken by shaking vigorously for 3 × 30 s with 5 min rest between pulses using a FastPrep-24 bead beating grinder (MP bio). After centrifuging for 10 min at 15,000$g$, supernatant was transferred to clean tubes and protein concentration was determined using the Protein A280 function of a NanoDrop 1000 (Thermo Fisher Scientific). The samples were diluted to equal concentrations of 2.5 mg/ml in a total volume of 0.5 ml, with 30 µl frozen as input sample. Beads were equilibrated by washing three times in 0.8 ml IP lysis buffer. Anti-HA-His Amintra NHS-activated Resin (MRC PPU Reagents & Services) was used for HA IP, with 25 µl slurry used per sample and centrifugation steps at 2,500$g$ for 1 min to isolate the resin. For FLAG IP, anti-FLAG M2 magnetic beads (M8823; Sigma-Aldrich) were used with ~20 µl slurry per IP and a magnetic rack to isolate the beads. To ensure pipetting consistency, beads/resin was resuspended in a 10-fold excess of IP buffer before addition to samples. Pulldown was performed overnight at 4°C on a tube carousel. Beads/resin with bound proteins were washed four times with 300 µl IP buffer, and elution for HA pulldown was performed in 60 µl 2× sample buffer (4% SDS, 20% glycerol, 10% 2-mercaptoethanol, 0.004% bromophenol blue, and 0.125 M Tris–HCl, pH 6.8) at 95°C for 10 min. For FLAG IP, elution was carried out at 4°C in 100 µl IP buffer containing 150 ng/µl 3xFLAG peptide (F4799; Sigma-Aldrich) before mixing with 2× sample buffer in a separate tube and boiling. The input samples were mixed with 2× sample buffer and boiled, and the samples were analysed by Western blotting.

### Phosphatase treatment experiment

Yeast strains were grown as duplicate 30 ml YEPD liquid cultures for 6 h at a starting density of 0.2 OD, with 200 nM rapamycin treatment added for the final 2 h. Cell pellets consisting of ~18 OD were washed in cold water and snap frozen on dry ice for storage at −20°C. Duplicate pellets were lysed in 0.5 ml of IP lysis buffer (lacking NEM) with or without PhosSTOP using glass beads as described for IP. Undiluted cleared lysates were set up as 50-µl reactions containing 1 mM MnCl2 final, and 1 µl of Λ protein phosphatase (P0753S; NEB) or IP lysis buffer was added to each reaction for the indicated time with incubation at 30°C. To stop the reaction, the samples were placed on ice, and an equal volume of 2× sample buffer was added

before boiling for 10 min. A volume of 10 µl per lane of each sample was analysed by Phos-tag Western blotting.

## Data Availability

All data, associated methods, and sources of materials are available in the main text or in the supporting information.

## Supplementary Information

## Acknowledgements

We thank Karim Labib's laboratory for the gift of plasmid pKL259 and use of equipment. Many thanks to SLS Central Technical Service (including Media Prep team), MRC PPU Reagents & Services, MRC PPU DNA Sequencing and Services, and MRC PPU Mass Spectrometry facility, laboratory management, and support staff for their assistance. This work was supported by the Medical Research Council (grant number MC_UU_00018/8 to A Rousseau).

### Author Contributions

A Agrotis: conceptualization, investigation, methodology, and writing—original draft, review, and editing.
F Lamoliatte: formal analysis and methodology.
TD Williams: investigation, methodology, and writing—review and editing.
A Black: investigation, methodology, and writing—review and editing.
R Horberry: investigation.
A Rousseau: conceptualization, supervision, funding acquisition, methodology, project administration, and writing—review and editing.

### Conflict of Interest Statement

The authors declare that they have no conflict of interest.

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
