## [Reviewer comments · Life Science Alliance]

Life Science Alliance

Multiple phosphorylation of Cdc48/p97 cofactor protein Shp1/p47 during cell stress in budding yeast

Alexander Agrotis, Frederic Lamoliatte, Thomas Williams, Ailsa Black, Rhuari Horberry, and Adrien Rousseau
DOI: <https://doi.org/10.26508/lsa.202201642>

Corresponding author(s): Adrien Rousseau, University of Dundee

Review Timeline:

Submission Date:	2022-08-01
Editorial Decision:	2022-09-08
Revision Received:	2022-12-09
Editorial Decision:	2023-01-13
Revision Received:	2023-01-16
Accepted:	2023-01-16

Scientific Editor: Novella Guidi

Transaction Report:

September 8, 2022

Re: Life Science Alliance manuscript #LSA-2022-01642

Dr. Adrien Rousseau
University of Dundee
MRC Protein Phosphorylation and Ubiquitylation Unit
School of Life Sciences
Dundee DD1 5EH
United Kingdom

Dear Dr. Rousseau,

Thank you for submitting your manuscript entitled "Double phosphorylation of Cdc48/p97 cofactor protein Shp1/p47 during cell stress in budding yeast" to Life Science Alliance. The manuscript was assessed by expert reviewers, whose comments are appended to this letter. We invite you to submit a revised manuscript addressing the Reviewer comments.

Thank you for this interesting contribution to Life Science Alliance. We are looking forward to receiving your revised manuscript.

Sincerely,

B. MANUSCRIPT ORGANIZATION AND FORMATTING:

Reviewer #1 (Comments to the Authors (Required)):

In the present manuscript, Agrotis et al. analyze the phosphorylation of the Cdc48 co-factor Shp1. The authors identify two sites which are increasingly phosphorylated in response to different stress conditions. They suggest that the identified phosphorylation is modulated by the kinase Mpk1 and the phosphatase Glc7. Several functions of Shp1 were analyzed, but none showed an effect that is due to altered Shp1 phosphorylation. The authors provide a solid identification and analysis of the Shp1 phosphorylation sites. The analysis of the involved kinase and phosphatase is still preliminary. Finally, the manuscript lacks any hint on the physiological relevance of the studied modification.

Major points:

1. In the abstract the authors state "Yeast engineered to lack these phosphorylation sites using CRISPR/Cas9 did not show defects in known Shp1 functions, suggesting these phosphorylations may instead have novel undiscovered functions." As the authors do not provide any evidence of a function of the studied phosphorylation site, this statement is not supported and should be removed from the abstract. As the authors state in the discussion, the identified phosphorylation sites might not have any function at all.
2. The authors nicely demonstrate phosphorylation of Shp1 under multiple stress conditions. Could the authors check if the phosphorylation deficient Shp1 mutant results in impaired growth under any of the conditions tested? This would provide a simple readout on the potential impact of the identified Shp1 phosphorylation sites.
3. In 5A the authors show Shp1 phosphorylation after 2h Cadmium treatment. In 5B and C the authors use only up to 20 min and see no effect of Cadmium. This is not comparable. Is Shp1 phosphorylation affected under this short treatment conditions? Would there be Shp1-phospho dependent effects on Met4 after 2h of Cadmium treatment?
4. The effects on Shp1 phosphorylation in dependence of Mpk1 as shown in Figs 7B, C and S4 are not convincing. In fact, the amount of double phosphorylated Shp1 appears highest in the cells expressing catalytic inactive Mpk1. As the authors base their conclusion on the ration of double-phosphorylated to unphosphorylated Shp1 a quantification of these bands might help to support their conclusion. Does Mpk1 bind to Shp1 or could these potential effects be indirect?
5. Figure 8 a and c: It is difficult to assess if the observed differences are due to impaired dephosphorylation. The authors should provide a normal SDS-PAGE of the same samples as shown in the blots, to analyze the steady state levels of Shp1. Does phosphorylation of Shp1 affect its stability?

Minor points

1. The authors state: "This observation highlights the importance of validating phosphorylation by experimental methods to confirm the relevant sites in vivo, compared to drawing early conclusions from phosphoproteomics data." The statement should be removed as the authors fail to demonstrate that the sites they study are functionally of any relevance.
2. In Figure 4E a Shp1 deletion strain should be added as a control, to show that the assay on autophagy function as performed in this manuscript shows the published Shp1 dependent effects.
3. Figure 8C: What is c1 and c2?

Reviewer #2 (Comments to the Authors (Required)):

The manuscript of Agrotis et al. addresses the important question how p97/Cdc48 cofactor proteins are posttranslationally modified after stress in yeast. In particular, the authors focus on the phosphorylation of the Shp1 adapter protein upon inhibition

of TORC1 by rapamycin. The authors identify two sites that are phosphorylated upon stress and identify Mpk1 as a kinase that is involved in this phosphorylation. Although the functional relevance of Shp1 phosphorylation remains unclear, the findings are relevant to the field and make a valuable contribution to the discussion on posttranslational modification of p97/Cdc48 adapter proteins.

While most of the conclusions in the manuscript are justified by the data presented, there are major points that need to be addressed before this manuscript is suitable for publication.

1. The main conclusions of the manuscript rely on the interpretation of band shifts in Phos-tag gels. The phosphorylation of Shp1 and the phosphosites are only shown indirectly and should also be shown directly using phospho mass spectrometry.

The highest band observed in the gels upon stress runs consistently higher than the highest band of the respective controls (e.g. Figs. 2B, 2C, 4D, 7C). However, the authors refer to both species as "double-phosphorylated Shp1" (e.g. p5), which is incorrect and confusing. The additional shift upon stress could for example indicate different phosphorylation sites before and after stress or additional phosphosite(s) that require(s) the phosphorylation of S108 and S315. The authors should clarify the identity of these bands.

The authors suggest the presence of four states of Shp1 (unphosphorylated, double phosphorylated, pS108 only, pS315 only). While this is true for the wild type yeast strains, the Shp1-5xFLAG strain consistently shows (at least) five different bands (Figs. 7C, 8C). The identity of the additional band(s) should be clarified experimentally.

2. The authors tested the effect of Shp1 phosphorylation on specific pathways that are known to involve Shp1. However, the authors should also assess general cell growth and survival of strains with wild type, the non-phosphorylatable mutant and a phosphomimic mutant of Shp1 upon stress. The phosphomimic mutant would also be an informative control for other experiments presented in this manuscript.

3. The authors show that deletion or inactivation of the kinase Mpk1 affects Shp1 phosphorylation and conclude that Mpk1 phosphorylates Shp1. However, this effect could also be indirect and the authors should show that Mpk1 directly phosphorylates Shp1 upon stress.

4. The mass spectrometry screen (Fig. 6) contains no useful information and does not contribute to the overall story of the paper. Figure 6 and the accompanying supplementary Fig. S3 should be removed from the manuscript, which will not diminish the significance of the paper.

5. The approach of glc7 removal using a temperature sensitive yeast strain has the conceptual problem that the elevated temperature causes an increase in Shp1 phosphorylation. Additionally, the cellular stress generated by Glc7/PP1 depletion is likely to induce Shp1 phosphorylation, since the authors show that a broad range of cellular stresses increase Shp1 phosphorylation. Thus, the approach chosen in this manuscript is not compatible with monitoring Shp1 phosphorylation and the authors cannot conclude that Glc7/PP1 has a role in Shp1 dephosphorylation. The authors should therefore remove Figure 8 from the manuscript.

In line with this, the functional connection of Glc7/PP1 and Shp1/p47 as well as their interaction is taken out of context and should be removed from the discussion section.

Additionally, there are some minor points that need to be clarified:

6. TORC1 inhibition by rapamycin seems to increase the protein level of Shp1 (e.g. Fig. 2A), which complicates the comparison of band intensities before and after stress. The authors should mention this effect in the manuscript and consider this in their interpretation.

7. Replacement of Mpk1 by the catalytically inactive mutant increases Shp1 protein levels compared to wild type Mpk1 (Fig. 7C). Do the authors have an explanation for the accumulation of Shp1?

8. The effect of Mpk1 deletion or inactivation (Fig. 7) seems to mainly lead to an accumulation of the non-phosphorylated form of Shp1. This could indicate that Mpk1 is important for the initial phosphorylation of Shp1, while other kinases perform subsequent addition of other phosphates.

We would like to thank the editors for their time and valuable remarks. As described hereafter, we have invested great efforts in order to improve the manuscript in light with the reviewers' comments. Thus, all points have been addressed separately in the point-by-point rebuttal.

This has been done.

This has been done.

This has been added: "This study shows that phosphorylation of the yeast p97/Cdc48 adaptor protein Shp1 is enhanced upon multiple stresses, a process regulated by both Mpk1 kinase and PP1 phosphatase."

Yes, we do.

B. MANUSCRIPT ORGANIZATION AND FORMATTING:

Full guidelines are available on our Instructions for Authors page, <https://www.life-science->

alliance.org/authors

Reviewer #1 (Comments to the Authors (Required)):

In the present manuscript, Agrotis et al. analyze the phosphorylation of the Cdc48 co-factor Shp1. The authors identify two sites which are increasingly phosphorylated in response to different stress conditions. They suggest that the identified phosphorylation is modulated by the kinase Mpk1 and the phosphatase Glc7. Several functions of Shp1 were analyzed, but none showed an effect that is due to altered Shp1 phosphorylation. The authors provide a solid identification and analysis of the Shp1 phosphorylation sites. The analysis of the involved kinase and phosphatase is still preliminary. Finally, the manuscript lacks any hint on the physiological relevance of the studied modification.

We thank the reviewer for his/her in-depth assessment of our manuscript and for providing constructive comments and suggestions. We addressed each of them in the revised manuscript. We direct the reviewer to our responses to each individual question below.

Major points:

1. In the abstract the authors state "Yeast engineered to lack these phosphorylation sites using CRISPR/Cas9 did not show defects in known Shp1 functions, suggesting these phosphorylations may instead have novel undiscovered functions."

As the authors do not provide any evidence of a function of the studied phosphorylation site, this statement is not supported and should be removed from the abstract. As the authors state in the discussion, the identified phosphorylation sites might not have any function at all.

We agree with the reviewer and the statement has been removed.

2. The authors nicely demonstrate phosphorylation of Shp1 under multiple stress conditions. Could the authors check if the phosphorylation deficient Shp1 mutant results in impaired growth under any of the conditions tested? This would provide a simple readout on the potential impact of the identified Shp1 phosphorylation sites.

We thank the reviewer for the suggestion and this experiment has been done. We now show that Shp1 phosphomutants have slight defects in rescuing the growth of *shp1Δ* cells upon rapamycin treatment (Figure S3B). This indicates that the function of Shp1, while affected, is not completely abolished by the loss of phosphorylation. This has been added in the main manuscript: “Moreover, Shp1 phosphomutants only showed slight defects in rescuing the growth of *shp1Δ* cells upon rapamycin treatment, indicating that the function of Shp1 is not completely abolished by the loss of phosphorylation (Figure S3B).”.

3. In 5A the authors show Shp1 phosphorylation after 2h Cadmium treatment. In 5B and C the authors use only up to 20 min and see no effect of Cadmium. This is not comparable. Is Shp1 phosphorylation affected under this short treatment conditions? Would there be Shp1-phospho dependent effects on Met4 after 2h of Cadmium treatment?

This is a fair comment and we have now tested Shp1 phosphorylation after 20 min by PhosTag gels and we now show that 10- and 20-min Cadmium treatment is not sufficient to induce Shp1 phosphorylation. This suggests that it could still have Shp1-phospho dependent effects on Met4 after 2h, when Shp1 is efficiently phosphorylated upon Cadmium treatment. We have performed this experiment and we now show that Met4 ubiquitination defect is not apparent anymore after treatment with Cadmium for 2h in *shp1Δ* cells indicating that Shp1 loss only slows-down the deubiquitination kinetic of Met4 upon Cadmium stress. As the timing of Shp1 phosphorylation is not correlated with Met4 ubiquitination, it is unlikely that it contributes to Met4 regulation by ubiquitin.

This has been added to the main text: “Moreover, after 2 hours treatment with cadmium, no difference in Met4 ubiquitination was observed in *shp1Δ* cells compared to their WT counterpart, indicating that Shp1 loss slows the deubiquitination kinetics of Met4, rather than abolishing it, upon cadmium stress (Figure 5D). These results, combined with our observation that Shp1 phosphorylation does not appear to increase within 20 minutes of cadmium stress (Figure 5E), suggest that Shp1 phosphorylation at S108 and S315 does not play an important role in SCF^{Met30} disassembly.”.

4. The effects on Shp1 phosphorylation in dependence of Mpk1 as shown in Figs 7B, C and S4 are not convincing. In fact, the amount of double phosphorylated Shp1 appears highest in the cells expressing catalytic inactive Mpk1. As the authors base their conclusion on the ration of double-phosphorylated to unphosphorylated Shp1 a quantification of these bands might help to support their conclusion. Does Mpk1 bind to Shp1 or could these potential effects be indirect?

We agree with the reviewer that quantification would have been nice but unfortunately reliable quantification is not possible due to difficulties in efficiently separating the different phosphorylated forms of Shp1, so quantification will not be accurate. However, we agree that the kinase-dead (KD) mutant is less convincing, and this could be due to remaining kinase activity. In agreement with that, it has been recently shown that this KD mutant is still able to rescue yeast stress sensitivity for a subset of stresses suggesting that the kinase activity may indeed not be completely abolished. Therefore, we decided to remove the data

using the KD by only showing the rescue experiment with WT Mpk1 after 1h and 3h rapamycin treatment (Figure 6C).

We also agree with the reviewer that Mpk1 may indirectly controls Shp1 phosphorylation, and this is now better reported in the manuscript: "In agreement with this, we have shown that Mpk1 is important for Shp1 phosphorylation at these two SPxP sites, although whether this is direct or indirect remains to be elucidated. Indeed, one could imagine that Mpk1 phosphorylates and hence activates another kinase phosphorylating Shp1 upon stress."

5. Figure 8 a and c: It is difficult to assess if the observed differences are due to impaired dephosphorylation. The authors should provide a normal SDS-PAGE of the same samples as shown in the blots, to analyze the steady state levels of Shp1. Does phosphorylation of Shp1 affect its stability?

We agree with the reviewer that adding a normal SDS-PAGE to monitor Shp1 levels in Glc7-12 ts mutants would be a good addition to the paper. In addition, to strengthen our data, we have now added a rescue experiment in which Glc7 phosphatase has been expressed in Glc7 ts mutants. These new results show that expressing WT Glc7 protein in Glc7-12 ts mutants (C1: clone 1 and C2: clone 2) reduces Shp1 phosphorylation to WT levels at the restrictive temperature (Figure 7D), while the total protein levels of Shp1 analysed by SDS-PAGE remain unaffected (Figure 7E). This confirms that Glc7 inhibition induces Shp1 phosphorylation without affecting the abundance, and hence the stability, of Shp1 protein.

This has been added in the main text: "Consistently, Glc7 inactivation further caused increased phosphorylation of endogenously tagged Shp1-5xFLAG (**Figure 7C**). This can be prevented by re-expressing WT Glc7-3xHA (**Figure 7D**). Moreover, Shp1 abundance remained unaffected by inhibition of Glc7 indicating that the regulation mainly occurs at the phosphorylation level (**Figure 7E**). Therefore, Glc7 is a likely regulator of Shp1 phosphorylation."

Minor points

1. The authors state: "This observation highlights the importance of validating phosphorylation by experimental methods to confirm the relevant sites in vivo, compared to drawing early conclusions from phosphoproteomics data." The statement should be removed as the authors fail to demonstrate that the sites they study are functionally of any relevance.

In this sentence, we meant that sites identified by phosphoproteomic are not always phosphorylated in vivo. This was the case for S321/322 and T331 of Shp1 for example, and we wanted to mention that it is important to perform mutagenesis to confirm that the site is also phosphorylated in vivo. We did not mean functionally relevant. However, this highlights the ambiguity of the sentence, we therefore decided to remove it, as suggested by the reviewer.

2. In Figure 4E a Shp1 deletion strain should be added as a control, to show that the assay on autophagy function as performed in this manuscript shows the published Shp1 dependent effects.

This is a fair point. We have now added the analysis of autophagy induction in *shp1Δ* cells upon rapamycin treatment and we observed that autophagy is indeed defective in *shp1Δ* cells compared to WT cells. We thank the reviewer for the suggestion and the text has been edited accordingly: "Following 2 hours of rapamycin treatment we detected a clear increase in free GFP accumulation in WT yeast but not autophagy-defective *atg1Δ* yeast, as expected (Figure 4E and 4F). Autophagy is also defective in *shp1Δ* yeast, as previously described. Endogenous Shp1 2SA phosphomutant strain did not display any noticeable defect in autophagy, suggesting that Shp1 phosphorylation is not required for autophagy."

3. Figure 8C: What is c1 and c2?

c1 and c2 are two different clones of Shp1-5xFLAG knock-in in the Glc7-12 background. This has been now specified in the figure legend: "c1: clone 1 and c2: clone 2."

Reviewer #2 (Comments to the Authors (Required)):

The manuscript of Agrotis et al. addresses the important question how p97/Cdc48 cofactor proteins are postranslationally modified after stress in yeast. In particular, the authors focus on the phosphorylation of the Shp1 adapter protein upon inhibition of TORC1 by rapamycin. The authors identify two sites that are phosphorylated upon stress and identify Mpk1 as a kinase that is involved in this phosphorylation. Although the functional relevance of Shp1 phosphorylation remains unclear, the findings are relevant to the field and make a valuable contribution to the discussion on posttranslational modification of p97/Cdc48 adapter proteins.

While most of the conclusions in the manuscript are justified by the data presented, there are major points that need to be addressed before this manuscript is suitable for publication.

We would like to thank the reviewer for valuable and constructive comments and suggestions. We have made substantial revision to the manuscript to address them. We direct the reviewer to our responses to each individual question below.

1. The main conclusions of the manuscript rely on the interpretation of band shifts in Phos-tag gels. The phosphorylation of Shp1 and the phosphosites are only shown indirectly and should also be shown directly using phospho mass spectrometry.

We agree with the reviewer point but this has already been done in the literature. Indeed, these two sites (along with other S/TP sites) have previously been identified as being phosphorylated in various phosphoproteomic datasets (MacGilvray ME, et al., 2020; Albuquerque CP, et al., 2008; Pultz D, et al., 2012; Swaney DL, et al., 2013; Ficarro SB, et al.,

2002; Rødkær SV, et al., 2014; Lanz MC, et al., 2021). As this is an important point, we now clearly mention that these sites have previously been identified by phosphoproteomic experiments and cite relevant literature. This has been added to the manuscript :“ Since SP/TP motifs are amongst those reported to be frequently phosphorylated by kinases acting downstream of TORC1 (21), we initially focused on testing the five residues corresponding to SP (Serine Proline) and TP (Threonine Proline) motifs found in Shp1 (S108, S315, S321, S322, T331), as they have been previously identify by mass spectrometry as being phosphorylated”.

The highest band observed in the gels upon stress runs consistently higher than the highest band of the respective controls (e.g. Figs. 2B, 2C, 4D, 7C). However, the authors refer to both species as "double-phosphorylated Shp1" (e.g. p5), which is incorrect and confusing. The additional shift upon stress could for example indicate different phosphorylation sites before and after stress or additional phosphosite(s) that require(s) the phosphorylation of S108 and S315. The authors should clarify the identity of these bands.

We agree with the reviewer that it seems to have 5 different bands suggesting that an additional phosphorylation site that require priming phosphorylation of S108 or S315 may be present. This has been clarified all along the manuscript. Moreover, we have now identified that S108 is priming the phosphorylation of S106 upon rapamycin treatment. This has been added to the manuscript: “It is worth noting that the band corresponding to phosphorylated Shp1 upon rapamycin treatment migrated consistently higher than the uppermost band of Shp1 at steady state. This additional shift is not present in the Shp1-S108A/S315A mutant, indicating that either S108 and/or S315 may act as a priming site for an additional phosphorylation upon stress. Moreover, Shp1-S315A, but not Shp1-S108A, migrated higher upon rapamycin treatment, indicating that S108 is likely responsible for the additional phosphorylation (**Figure 3B**). Interestingly, peptide harbouring S108 has been shown to be dually phosphorylated at S106 and S108 by mass spectrometry (18, 20, 22, 23), suggesting that S108 could prime the phosphorylation at S106. Analysing the impact of S106 mutation, we observed that the higher migrating band was lost in Shp1-S106A upon rapamycin treatment compared to its WT counterpart (**Figure S2**). Moreover, S106A mutation prevented rapamycin-induced phosphorylation of Shp1-S315A (**Figure 3B and S2**). These results show that S108 is likely a priming site for S106. Overall, our data indicates that yeast Shp1 is phosphorylated at either of the two residues S108 and S315 under non-stressed conditions but, following stress, the majority of Shp1 becomes phosphorylated at both residues with S108A being important for the additional phosphorylation of S106.”.

We thank the reviewer for highlighting this point, this is a great addition to the paper.

The authors suggest the presence of four states of Shp1 (unphosphorylated, double phosphorylated, pS108 only, pS315 only). While this is true for the wild type yeast strains, the Shp1-5xFLAG strain consistently shows (at least) five different bands (Figs. 7C, 8C). The identity of the additional band(s) should be clarified experimentally.

We agree with the reviewer and this point has been address in the previous response.

2. The authors tested the effect of Shp1 phosphorylation on specific pathways that are

known to involve Shp1. However, the authors should also assess general cell growth and survival of strains with wild type, the non-phosphorylatable mutant and a phosphomimic mutant of Shp1 upon stress. The phosphomimic mutant would also be an informative control for other experiments presented in this manuscript.

We thank the reviewer for the suggestion and this experiment has now been done. We now show that Shp1 phosphomutants have slight defects in rescuing the growth of *shp1Δ* cells upon rapamycin treatment (Figure S3B). This indicates that the function of Shp1, while affected, is not completely abolished by the loss of phosphorylation. This has been added in the main manuscript: “Moreover, Shp1 phosphomutants only showed slight defects in rescuing the growth of *shp1Δ* cells upon rapamycin treatment, indicating that the function of Shp1 is not completely abolished by the loss of phosphorylation (Figure S3B).”.

As the phosphomimetic mutation (1 negative charge) is not recapitulating phosphorylation (2 negative charges), we usually prefer not to use it. Even if there is an effect, we would not be able to conclude that this is due to the phosphorylation.

3. The authors show that deletion or inactivation of the kinase Mpk1 affects Shp1 phosphorylation and conclude that Mpk1 phosphorylates Shp1. However, this effect could also be indirect and the authors should show that Mpk1 directly phosphorylates Shp1 upon stress.

This is a good suggestion. We have tested that, and we found that incubation of yeast recombinant Mpk1-6xHis with bacterial recombinant Shp1-6xHis was not able to induce Shp1 phosphorylation *in vitro* (see below). We stressed cells expressing Mpk1 before extraction and purification but looking at Mpk1 on PhosTag gel we noticed that only a very weak band showed a higher shift which correspond to activated Mpk1 phosphorylated on its TEY motif by Mkk1/2. Therefore, we think that Mpk1 is losing almost all activity during extraction and purification making this assay not reliable. Due to this issue, we would prefer not to include this data until efficient purification of active Mpk1 is achieved.

Nonetheless, we now clearly state in the manuscript that identifying whether Mpk1 is directly phosphorylating Shp1 will be important in the future: “In agreement with this, we have shown that Mpk1 is important for Shp1 phosphorylation at these two SPxP sites, although whether this is direct or indirect remains to be elucidated. Indeed, one could imagine that Mpk1 phosphorylates and hence activates another kinase phosphorylating Shp1 upon stress.”.

4. The mass spectrometry screen (Fig. 6) contains no useful information and does not contribute to the overall story of the paper. Figure 6 and the accompanying supplementary Fig. S3 should be removed from the manuscript, which will not diminish the significance of the paper.

We agree with the reviewer that removing this figure may help the flow of the paper. This figure and the associated text have been removed in the new version of the manuscript.

5. The approach of glc7 removal using a temperature sensitive yeast strain has the conceptual problem that the elevated temperature causes an increase in Shp1 phosphorylation. Additionally, the cellular stress generated by Glc7/PP1 depletion is likely to induce Shp1 phosphorylation, since the authors show that a broad range of cellular stresses increase Shp1 phosphorylation. Thus, the approach chosen in this manuscript is not compatible with monitoring Shp1 phosphorylation and the authors cannot conclude that Glc7/PP1 has a role in Shp1 dephosphorylation. The authors should therefore remove Figure 8 from the manuscript.

In line with this, the functional connection of Glc7/PP1 and Shp1/p47 as well as their interaction is taken out of context and should be removed from the discussion section.

We agree with the reviewer that the depletion of Glc7/PP1 may induce a stress which is responsible for Shp1 phosphorylation, and therefore that Glc7 indirectly controls Shp1 phosphorylation. Nonetheless, we do think that the Glc7 dataset is important, as Shp1 is already known to regulate Glc7 complex assembly, and hence Glc7 could dephosphorylate Shp1 as a feedback loop mechanism.

To address the reviewer's concern about indirect effect, we know clearly state it in the main text to prevent any ambiguity: "Therefore, Glc7 is a likely regulator of Shp1 phosphorylation. Whether Glc7 directly dephosphorylates Shp1, or its loss induced a stress that is indirectly triggering Shp1 phosphorylation, remains to be determined.", and "An alternative scenario would be that Glc7 loss triggers a stress activating a kinase phosphorylating Shp1. Further studies will help in understanding the reciprocal regulation between Shp1 and Glc7 complex."

Additionally, there are some minor points that need to be clarified:

6. TORC1 inhibition by rapamycin seems to increase the protein level of Shp1 (e.g. Fig. 2A), which complicates the comparison of band intensities before and after stress. The authors should mention this effect in the manuscript and consider this in their interpretation.

We showed in Figure 1 that the total level of Shp1 is not affected by rapamycin treatment using SDS-PAGE. PhosTag gels are not suited for looking at total protein levels, as the bands corresponding to unphosphorylated Shp1 and Shp1 harbouring single phosphorylation are converging toward the most phosphorylated form of Shp1 upon stress, making its signal much stronger, possibly through avidity effect. To prevent any confusion, SDS-PAGE analysis of WT and 2SA Shp1-5xFlag under unstressed and stressed conditions has been added to the new version of the manuscript (Figure S3A): "The total levels of WT and 2SA Shp1-5xFlag remained unchanged in both unstressed and stressed conditions indicating that Shp1 phosphorylation is not regulating Shp1 level (Figure S3A)".

7. Replacement of Mpk1 by the catalytically inactive mutant increases Shp1 protein levels compared to wild type Mpk1 (Fig. 7C). Do the authors have an explanation for the accumulation of Shp1?

We agree with the reviewer that the kinase-dead (KD) mutant may affect Shp1 level possibly by stabilising it after binding. Moreover, it has been recently shown that this KD mutant is still able to rescue yeast stress sensitivity for a subset of stresses suggesting that the kinase activity may not be completely abolished. Therefore, we decided to remove the data using the KD by only showing the rescue experiment with WT Mpk1 after 1h and 3h rapamycin treatment (Figure 6C) to prevent any wrong interpretation.

8. The effect of Mpk1 deletion or inactivation (Fig. 7) seems to mainly lead to an accumulation of the non-phosphorylated form of Shp1. This could indicate that Mpk1 is important for the initial phosphorylation of Shp1, while other kinases perform subsequent addition of other phosphates.

This is a very interesting suggestion and, as such, this has now been added to the manuscript: "Moreover, Mpk1 loss mainly leads to an accumulation of the non-phosphorylated form of Shp1. This could indicate that Mpk1 is important for the initial phosphorylation of Shp1, while other kinases perform subsequent addition of other phosphates."

January 13, 2023

RE: Life Science Alliance Manuscript #LSA-2022-01642R

Dr. Adrien Rousseau
University of Dundee
MRC Protein Phosphorylation and Ubiquitylation Unit
School of Life Sciences
Dundee DD1 5EH
United Kingdom

Dear Dr. Rousseau,

Thank you for submitting your revised manuscript entitled "Multiple phosphorylation of Cdc48/p97 cofactor protein Shp1/p47 during cell stress in budding yeast". We would be happy to publish your paper in Life Science Alliance pending final revisions necessary to meet our formatting guidelines.

- please upload both your main and supplementary figures as single files
- please make sure that the author list in the manuscript and the author list in our system match
- please add the Author Contributions to the main manuscript text
- please use the [10 author names, et al.] format in your references (i.e. limit the author names to the first 10)
- please add your Figure 8 Figure legend to the main manuscript text and add a new file for Figure 8 (the page is currently blank)
- please incorporate your supp. material file into the main manuscript

Figure Check:

- please add sizes next to all blots

A. FINAL FILES:

B. MANUSCRIPT ORGANIZATION AND FORMATTING:

Sincerely,

Reviewer #1 (Comments to the Authors (Required)):

The authors have sufficiently addressed all points raised by the reviewers.

Reviewer #2 (Comments to the Authors (Required)):

The authors have satisfactorily addressed most of my concerns with the previous manuscript. The revised manuscript is ready for publication.

January 16, 2023

RE: Life Science Alliance Manuscript #LSA-2022-01642RR

Dr. Adrien Rousseau
University of Dundee
MRC Protein Phosphorylation and Ubiquitylation Unit
School of Life Sciences
Dundee DD1 5EH
United Kingdom

Dear Dr. Rousseau,

Thank you for submitting your Research Article entitled "Multiple phosphorylation of Cdc48/p97 cofactor protein Shp1/p47 during cell stress in budding yeast". It is a pleasure to let you know that your manuscript is now accepted for publication in Life Science Alliance. Congratulations on this interesting work.

DISTRIBUTION OF MATERIALS:

Again, congratulations on a very nice paper. I hope you found the review process to be constructive and are pleased with how the manuscript was handled editorially. We look forward to future exciting submissions from your lab.

Sincerely,
